# Temporal Knowledge Graph Forecasting
# Without Knowledge Using In-Context Learning

**Dong-Ho Lee***,  **Kian Ahrabian***,  **Woojeong Jin,  Fred Morstatter,  Jay Pujara**

Department of Computer Science and Information Sciences Institute
University of Southern California
{dongho.lee,ahrabian,woojeong.jin}@usc.edu, {fredmors,jpujara}@isi.edu

## Abstract

Temporal knowledge graph (TKG) forecasting benchmarks challenge models to predict future facts using knowledge of past facts. In this paper, we develop an approach to use in-context learning (ICL) with large language models (LLMs) for TKG forecasting. Our extensive evaluation compares diverse baselines, including both simple heuristics and state-of-the-art (SOTA) supervised models, against pre-trained LLMs across several popular benchmarks and experimental settings. We observe that naive LLMs perform on par with SOTA models, which employ carefully designed architectures and supervised training for the forecasting task, falling within the (-3.6%, +1.5%) Hits@1 margin relative to the median performance. To better understand the strengths of LLMs for forecasting, we explore different approaches for selecting historical facts, constructing prompts, controlling information propagation, and parsing outputs into a probability distribution. A surprising finding from our experiments is that LLM performance endures ($\pm$0.4% Hit@1) even when semantic information is removed by mapping entities/relations to arbitrary numbers, suggesting that prior semantic knowledge is unnecessary; rather, LLMs can leverage the symbolic patterns in the context to achieve such a strong performance. Our analysis also reveals that ICL enables LLMs to learn irregular patterns from the historical context, going beyond frequency and recency biases[1].

## 1   Introduction

Knowledge Graphs (KGs) are prevalent resources for representing real-world facts in a structured way. While traditionally, KGs have been utilized for representing static snapshots of "current" knowledge, recently, temporal KGs (TKGs) have gained popularity to preserve the complex temporal dynamics

---

*Authors contributed equally.

[1]https://github.com/usc-isi-i2/isi-tkg-icl

---

**Which team will win the Super Bowl in 2023?**

```
2000: [Superbowl, Champion, St Louis]
2001: [Superbowl, Champion, Baltimore]
2002: [Superbowl, Champion, New England]
2003: [Superbowl, Champion, Tampa Bay]
...
2019: [Superbowl, Champion, New England]
2020: [Superbowl, Champion, Kansas City]
2021: [Superbowl, Champion, Tampa Bay]
2022: [Superbowl, Champion, Los Angeles]
2023: [Superbowl, Champion,
```

| Model | # Params. | Prediction | G.T Rank |
|---|---|---|---|
| EleutherAI gpt-j-6b | 6B | Los Angeles | 3 |
| EleutherAI gpt-neox-20b | 20B | Kansas City | 1 |
| OpenAI text-ada-001 | 350M | Carolina | >5 |
| OpenAI text-babbage-001 | 1.3B | Indianapolis | >5 |
| OpenAI text-curie-001 | 6.7B | New England | >5 |
| OpenAI text-davinci-003 | 175B | TBD | >5 |
| OpenAI gpt-3.5-turbo | - | Sorry, I cannot predict future events. | |

Table 1: **Model predictions for an example structured prompt**. Given a structured prompt (*top*), large language models forecast the missing fact using next token prediction (*bottom*). G.T Rank indicates the rank of the ground truth response ("*Kansas City*") within the token probability distribution.

of knowledge (Leblay and Chekol, 2018; García-Durán et al., 2018; Goel et al., 2020; Lacroix et al., 2020). Recent endeavors in the area of TKGs have been focused on predicting future missing links (*i.e.,* forecasting), given a query quadruple $q = (s_i, p_j, ?, t_T)$ and a set of associated historical facts $\mathcal{E}_q = \{(s, p, o, t) \,|\, t < t_T\} \subseteq \mathcal{E}$ (Gastinger et al., 2022). An illustrative example of this task is the question "*Which team will win the Super Bowl in 2023?*" that can be expressed as $q = (Super\ bowl, Champion, ?, 2023)$ and $\mathcal{E}_q = \{(Super\ bowl, Champion, Los\ Angeles, 2022), (Super\ bowl, Champion, Tampa\ Bay, 2021), ...\}$ (See Table 1). The ultimate objective is to identify the most suitable entity among all the football teams $\mathcal{E}$ (*e.g., St Louis, Baltimore*) to fill the missing field.

Prior research on TKG forecasting has been primarily focused on developing supervised ap-

proaches such as employing graph neural networks to model interrelationships among entities and relations (Jin et al., 2020; Li et al., 2021; Han et al., 2021b,a), using reinforcement learning techniques (Sun et al., 2021), and utilizing logical rules (Zhu et al., 2021; Liu et al., 2022). However, these techniques have prominent limitations, including the need for large amounts of training data that include thorough historical information for the entities. Additionally, model selection is a computationally expensive challenge as the state-of-the-art approach differs for each dataset.

In this paper, we develop a TKG forecasting approach by casting the task as an in-context learning (ICL) problem using large language models (LLMs). ICL refers to the capability of LLMs to learn and perform an unseen task efficiently when provided with a few examples of input-label pairs in the prompt (Brown et al., 2020). Prior works on ICL usually leverage few-shot demonstrations, where a uniform number of examples are provided for each label to solve a classification task (Min et al., 2022; Wei et al., 2023). In contrast, our work investigates what the model learns from irregular patterns of historical facts in the context. We design a three-stage pipeline to control (1) the background knowledge selected for context, (2) the prompting strategy for forecasting, and (3) decoding the output into a prediction. The first stage uses the prediction query to retrieve a set of relevant past facts from the TKG that can be used as context (Section 3.1). The second stage transforms these contextual facts into a lexical prompt representing the prediction task (Section 3.3). The third stage decodes the output of the LLM into a probability distribution over the entities and generates a response to the prediction query (Section 3.4). Our experimental evaluation performs competitively across a diverse collection of TKG benchmarks without requiring the time-consuming supervised training, or custom-designed architectures.

We present extensive experimental results on common TKG benchmark datasets such as WIKI (Leblay and Chekol, 2018), YAGO (Mahdisoltani et al., 2014), and ICEWS (García-Durán et al., 2018; Jin et al., 2020). Our findings are as follows: (1) LLMs demonstrate the ability to make predictions about future facts using ICL without requiring any additional training. Moreover, these models show comparable performance to supervised approaches, falling within the (-3.6%,

+1.5%) Hits@1 margin, relative to the median approach for each dataset; (2) LLMs perform almost identically when we replace entities' and relations' lexical names with numerically mapped indices, suggesting that the prior semantic knowledge is not a critical factor for achieving such a high performance; and (3) LLMs outperform the best heuristic rule-based baseline on each dataset (*i.e.,* the most frequent or the most recent, given the historical context) by (+10%, +28%) Hits@1 relative margin, indicating that they do not simply select the output using frequency or recency biases in ICL (Zhao et al., 2021).

## 2 Problem Formulation

**In-Context Learning.** ICL is an emergent capability of LLMs that aims to induce a state in the model to perform a task by utilizing contextual input-label examples, without requiring changes to its internal parameters (Brown et al., 2020). Formally, in ICL for classification, a prompt is constructed by linearizing a few input-output pair examples $(\mathbf{x}_i, \mathbf{y}_i)$ from the training data. Subsequently, when a new test input text $\mathbf{x}_{\text{test}}$ is provided, ICL generates the output $\mathbf{y}_{\text{test}} \sim \mathcal{P}_{\text{LLM}}(\mathbf{y}_{\text{test}} \mid \mathbf{x}_1, \mathbf{y}_1, \dots, \mathbf{x}_k, \mathbf{y}_k, \mathbf{x}_{\text{test}})$ where $\sim$ refers to decoding strategy.

**Temporal Knowledge Graph Forecasting.** Formally, a TKG, $\mathcal{G} = (\mathcal{V}, \mathcal{R}, \mathcal{E}, \mathcal{T})$, is comprised of a set of entities $\mathcal{V}$, relations $\mathcal{R}$, facts $\mathcal{E}$, and timestamps $\mathcal{T}$. Moreover, since time is sequential, $\mathcal{G}$ can be split into a sequence of time-stamped snapshots, $\mathcal{G} = \{\mathcal{G}_1, \mathcal{G}_2, \dots, \mathcal{G}_t, \dots\}$, where each snapshot, $\mathcal{G}_t = (\mathcal{V}, \mathcal{R}, \mathcal{E}_t)$, contains the facts at a specific point in time $t$. Each fact $f \in \mathcal{E}_t$ is a quadruple $(s, p, o, t)$ where $s, o \in \mathcal{V}$, $p \in \mathcal{R}$, and $t \in \mathcal{T}$. The TKG forecasting task involves predicting a temporally conditioned missing entity in the future given a query quadruple, $(?, p, o, t)$ or $(s, p, ?, t)$, and previous graph snapshots $\mathcal{G}_{1:t-1} = \{\mathcal{G}_1, \mathcal{G}_2, \dots, \mathcal{G}_{t-1}\}$. Here, the prediction typically involves ranking each entity's assigned score.

## 3 In-context Learning for Temporal Knowledge Graph Forecasting

In this work, we focus on 1) modeling appropriate history $\mathcal{E}_q$ for a given query quadruple $q$, 2) converting $\{\mathcal{E}_q, q\}$ into a prompt $\theta_q$, and 3) employing ICL to get prediction $\mathbf{y}_q \sim \mathcal{P}_{\text{LLM}}(\mathbf{y}_q \mid \theta_q)$ in a zero-shot manner. Here, the history $\mathcal{E}_q$ is modeled on the facts from the previous graph snapshots

$\mathcal{G}_{1:t-1} = \{\mathcal{G}_1, \mathcal{G}_2, \ldots, \mathcal{G}_{t-1}\}$, and we employ token probabilities for $\mathbf{y}_q$ to get ranked scores of candidate entities in a zero-shot manner. In the rest of this section, we study history modeling strategies (Sec 3.1), response generation approaches (Sec 3.2), prompt construction templates (Sec 3.3), and common prediction settings (Sec 3.4).

## 3.1 History Modeling

To model the history $\mathcal{E}_q$, we filter facts that the known entity or relation in the query $q$ has been involved in. Specifically, given the query quadruple $q = (s, p, ?, t)$ under the object entity prediction setting, we experiment with two different aspects of historical facts:

**Entity vs. Pair.** `Entity` includes past facts that contain $s$, *e.g.,* all historical facts related to *Superbowl*. In contrast, `Pair` includes past facts that contain both $s$ and $p$, *e.g.,* a list of (*Superbowl, Champion, Year*) as shown in Table 1.

**Unidirectional vs. Bidirectional.** `Unidirectional` includes past facts $\mathcal{F}$ wherein $s$ (`Entity`) or $(s, p)$ (`Pair`) is in the same position as it is in $q$ (*e.g.,* `Unidirectional` & `Pair` – $s$ and $p$ served as subject and predicate in $f \in \mathcal{F}$). `Bidirectional` includes past facts $\mathcal{F}$ wherein $s$ (`Entity`) or $(s, p)$ (`Pair`) appear in any valid position (*e.g.,* `Bidirectional` & `Entity` – $s$ served as subject or object in $f \in \mathcal{F}$). As an example of the `Bidirectional` setting, given $q = $ (*Superbowl, Champion, ?, 2023*), we include $f = $ (*Kupp, Played, Superbowl, 2022*) because $s$ (*i.e., Superbowl*) is present as the object in $f$. Moreover, in the `Bidirectional` setting, to preserve the semantics of the facts in the $\mathcal{E}_q$, we transform the facts where $s$ appears as an object by 1) swapping the object and subject and 2) replacing the relation with its uniquely defined inverse relation (*e.g.,* $(f_s, f_p, f_o, f_t) \rightarrow (f_o, f_p^{-1}, f_s, f_t)$).

## 3.2 Response Generation

Given a prompt $\theta_q$, we pass it to an LLM to obtain the next token probabilities. Then, we use the obtained probabilities to get a ranked list of entities. However, obtaining scores for entities based on these probabilities is challenging as they may be composed of several tokens. To address this challenge, we utilize a mapped numerical label as an indirect logit to estimate their probabilities (Lin et al., 2022).

| Category | Prompt |
|---|---|
| Lexical $\mathcal{L}(\cdot)$ | 2000: [Superbowl, Champion, 0. St Louis] 2001: [Superbowl, Champion, 1. Baltimore] … 2023: [Superbowl, Champion, |
| Index $\mathcal{I}(\cdot)$ | 2000: [0, 0, 0. 0] 2001: [0, 0, 1. 1] … 2023: [0, 0, |

Table 2: **Prompt Example.**

## 3.3 Prompt Construction

Given the history $\mathcal{E}_q$ and query $q$, we construct a prompt using a pre-defined template $\theta$. Specifically, given the query quadruple $q = (s, p, ?, t)$ under the object entity prediction setting, we present two versions of the template $\theta$ with varying levels of information. Our assumption is that each entity or relation has an indexed $\mathcal{I}(\cdot)$ (*e.g.,* 0) and a lexical $\mathcal{L}(\cdot)$ (*e.g., Superbowl*) form (See Table 2).

**Index.** `Index` displays every fact, $(f_s, f_p, f_o, f_t) \in \mathcal{E}$ using the "$f_t$:$[\mathcal{I}(f_s), \mathcal{I}(f_p), n_{f_o}. \mathcal{I}(f_o)]$" template where $f_s, f_o \in \mathcal{V}$, $f_p \in \mathcal{R}$, $f_t \in \mathcal{T}$, $n_{f_o}$ denotes an incrementally assigned numerical label (*i.e.,* indirect logit), and $\mathcal{I}$ is a mapping from entities to unique indices. For example, in Table 1, we can use the following mappings are for the entities and relations, respectively: {*Superbowl* $\rightarrow$ 0, *St Louis* $\rightarrow$ 1, *Baltimore* $\rightarrow$ 2} and {*Champion* $\rightarrow$ 0}. The query $q$ is then represented as "$t$:$[\mathcal{I}(s), \mathcal{I}(p),$", concatenated to the end of the prompt. For subject entity prediction, we follow the same procedure from the other side.

**Lexical.** `Lexical` follows the same process as `Index` but uses lexical form $\mathcal{L}(\cdot)$ of entity and relation. Each fact in $(f_s, f_p, f_o, f_t) \in \mathcal{E}$ is represented as "$f_t$:$[\mathcal{L}(f_s), \mathcal{L}(f_p), n_{f_o}. \mathcal{L}(f_o)]$" and the query $q$ is represented as "$t$:$[\mathcal{L}(s), \mathcal{L}(p),$", concatenated to the end of the prompt.

## 3.4 Prediction Setting

All the historical facts in the dataset are split into three subsets, $\mathcal{D}_{\text{train}}$, $\mathcal{D}_{\text{valid}}$, and $\mathcal{D}_{\text{test}}$, based on the chronological order with train $<$ valid $<$ test. Given this split, during the evaluation phase, the TKG forecasting task requires models to predict over $\mathcal{D}_{\text{test}}$ under the following two settings:

**Single Step.** In this setting, for each test query, the model is provided with ground truth facts from past timestamps in the *test* period. Hence, after making predictions for a test query in a specific

| Dataset | # Ents | # Rels | # of Facts | | | Interval |
|---------|--------|--------|-------|-------|------|----------|
| | | | Train | Valid | Test | |
| WIKI | 12,554 | 24 | 539,286 | 67,538 | 63,110 | 1 year |
| YAGO | 10,623 | 10 | 161,540 | 19,523 | 20,026 | 1 year |
| ICEWS14 | 6,869 | 230 | 74,845 | 8,514 | 7,371 | 1 day |
| ICEWS18 | 23,033 | 256 | 373,018 | 45,995 | 49,995 | 1 day |
| ACLED-CD22 | 243 | 6 | 1,788 | 216 | 222 | 1 day |

Table 3: **Data statistics.** Each dataset consists of entities, relations, and historical facts, with the facts within the same time interval identified by the same timestamp. The facts are divided into three subsets based on time, where train < valid < test.

timestamp, the ground truth fact for that query is added to the history before moving to the test queries in the next timestamp.

**Multi Step.** In this setting, the model is *not* provided with ground truth facts from past timestamps in the *test* period and has to rely on its noisy predictions. Hence, after making predictions for a test query in a specific timestamp, instead of the ground truth fact for that query, we add the predicted response to the history before moving to the test queries in the next timestamp. This setting is considered more difficult as the model is forced to rely on its own noisy predictions, which can lead to greater uncertainty with each successive timestamp.

## 4 Experimental Setup

### 4.1 Datasets

For our experiments, we use the WIKI (Leblay and Chekol, 2018), YAGO (Mahdisoltani et al., 2014), ICEWS14 (García-Durán et al., 2018), and ICEWS18 (Jin et al., 2020) benchmark datasets with the unified splits introduced in previous studies (Gastinger et al., 2022). Additionally, we extract a new temporal forecasting dataset from the Armed Conflict Location & Event Data Project (ACLED) project[2] which provides factual data of crises in a particular region. We specifically focus on incidents of combat and violence against civilians in Cabo Delgado from January 1900 to March 2022, using data from October 2021 to March 2022 as our test set. This dataset aims to investigate whether LLMs leverage prior semantic knowledge to make predictions and how effective they are when deployed in real-world applications. Table 3 presents the statistics of these datasets.

[2]https://data.humdata.org/organization/acled

| Model Family | Model Name | # Params | Instruction-tuned |
|--------------|------------|----------|-------------------|
| GPT2 | gpt2 | 124M | ✗ |
| | gpt2-medium | 355M | ✗ |
| | gpt2-large | 774M | ✗ |
| | gpt2-xl | 1.5B | ✗ |
| GPT-J | gpt-j-6b | 6B | ✗ |
| GPT-NeoX | gpt-neox-20b | 20B | ✗ |
| InstructGPT | gpt-3.5-turbo | - | ✓ |

Table 4: **Language Models used in the paper.** Exact model size of gpt-3.5-turbo is unknown.

### 4.2 Evaluation

We evaluate the models on well-known metrics for link prediction: Hits@$k$, with $k = 1, 3, 10$. Following (Gastinger et al., 2022), we report our results in two evaluation settings: 1) **Raw** retrieves the sorted scores of candidate entities for a given query quadruple and calculates the rank of the correct entity; and 2) **Time-aware filter** also retrieves the sorted scores but removes the entities that are valid predictions before calculating the rank, preventing them from being considered errors. To illustrate, if the test query is (NBA, Clinch Playoff, ?, 2023) and the true answer is Los Angeles Lakers, there may exist other valid predictions such as (NBA, Clinch Playoff, Milwaukee Bucks, 2023) or (NBA, Clinch Playoff, Boston Celtics, 2023). In such cases, the time-aware filter removes these valid predictions, allowing for accurate determination of the rank of the "Los Angeles Lakers." In this paper, we present performance with the time-aware filter.

### 4.3 Models.

As shown in Table 4, we perform experiments on four language model families. Among those, three are open-sourced: GPT2 (Radford et al., 2019), GPT-J (Wang, 2021), and GPT-NeoX (Black et al., 2022). All models employ the GPT-2 byte level BPE tokenizer (Radford et al., 2019) with nearly identical vocabulary size. In addition, we use the gpt-3.5-turbo model to analyze the performance of the instruction-tuned models. However, we do not directly compare this model to other models in terms of size since the actual model size is unknown. As for the TKG baselines, (*i.e.,* RE-Net (Jin et al., 2020), RE-GCN (Li et al., 2021), TANGO (Han et al., 2021b), xERTE (Han et al., 2021a), TimeTraveler (Sun et al., 2021), CyGNet (Zhu et al., 2021), and TLogic (Liu et al., 2022)), we report the numbers presented in prior research (Gastinger et al., 2022). Appendix A.4 provides more details on baseline models.

| Single-Step | Train | YAGO | | | WIKI | | | ICEWS14 | | | ICEWS18 | | | ACLED-CD22 | | |
|---|---|---|---|---|---|---|---|---|---|---|---|---|---|---|---|---|
| | | H@1 | H@3 | H@10 | H@1 | H@3 | H@10 | H@1 | H@3 | H@10 | H@1 | H@3 | H@10 | H@1 | H@3 | H@10 |
| RE-GCN | ✓ | 0.787 | 0.842 | 0.884 | 0.747 | 0.817 | 0.846 | 0.313 | 0.473 | **0.626** | **0.223** | **0.367** | **0.525** | **0.446** | **0.545** | **0.608** |
| xERTE | ✓ | 0.842 | 0.902 | **0.912** | 0.703 | 0.785 | 0.801 | 0.330 | 0.454 | 0.570 | 0.209 | 0.335 | 0.462 | 0.320 | 0.445 | 0.497 |
| TLogic | ✓ | 0.740 | 0.789 | 0.791 | **0.786** | **0.860** | **0.870** | **0.332** | 0.476 | 0.602 | 0.204 | 0.336 | 0.480 | 0.009 | 0.045 | 0.094 |
| TANGO | ✓ | 0.590 | 0.646 | 0.677 | 0.483 | 0.514 | 0.527 | 0.272 | 0.408 | 0.550 | 0.191 | 0.318 | 0.462 | 0.327 | 0.482 | 0.599 |
| Timetraveler | ✓ | **0.845** | **0.908** | **0.912** | 0.751 | 0.820 | 0.830 | 0.319 | 0.454 | 0.575 | 0.212 | 0.325 | 0.439 | 0.240 | 0.315 | 0.457 |
| GPT-NeoX (Entity) | ✗ | 0.784 | 0.891 | **0.927** | 0.694 | 0.804 | 0.844 | **0.324** | **0.460** | **0.565** | 0.192 | **0.313** | **0.414** | **0.324** | **0.492** | **0.604** |
| GPT-NeoX (Pair) | ✗ | **0.787** | **0.892** | 0.926 | **0.721** | **0.812** | **0.847** | 0.297 | 0.408 | 0.482 | **0.196** | 0.307 | 0.402 | 0.317 | 0.440 | 0.566 |

| Multi-Step | Train | YAGO | | | WIKI | | | ICEWS14 | | | ICEWS18 | | | ACLED-CD22 | | |
|---|---|---|---|---|---|---|---|---|---|---|---|---|---|---|---|---|
| | | H@1 | H@3 | H@10 | H@1 | H@3 | H@10 | H@1 | H@3 | H@10 | H@1 | H@3 | H@10 | H@1 | H@3 | H@10 |
| RE-GCN | ✓ | **0.717** | **0.776** | 0.817 | 0.594 | 0.648 | 0.678 | **0.278** | **0.421** | **0.575** | **0.195** | **0.326** | **0.475** | **0.421** | 0.464 | 0.502 |
| RE-Net | ✓ | 0.534 | 0.613 | 0.662 | 0.472 | 0.507 | 0.530 | **0.278** | 0.408 | 0.549 | 0.184 | 0.314 | 0.461 | 0.238 | 0.445 | 0.563 |
| CyGNet | ✓ | 0.613 | 0.742 | **0.834** | 0.525 | 0.624 | 0.675 | 0.266 | 0.402 | 0.545 | 0.166 | 0.295 | 0.444 | 0.408 | **0.500** | **0.588** |
| TLogic | ✓ | 0.631 | 0.706 | 0.715 | **0.613** | **0.663** | **0.682** | 0.265 | 0.395 | 0.531 | 0.155 | 0.272 | 0.412 | 0.009 | 0.045 | 0.094 |
| GPT-NeoX (Entity) | ✗ | 0.686 | **0.793** | **0.840** | 0.543 | 0.622 | **0.655** | **0.247** | **0.363** | **0.471** | 0.136 | 0.224 | 0.321 | **0.319** | **0.417** | **0.500** |
| GPT-NeoX (Pair) | ✗ | **0.688** | **0.793** | 0.839 | **0.570** | **0.625** | 0.652 | 0.236 | 0.324 | 0.395 | **0.155** | **0.245** | **0.331** | 0.289 | 0.410 | 0.464 |

Table 5: **Performance (Hits@K)** comparison between supervised models and ICL for single-step (top) and multi-step (bottom) prediction. The first group in each table consists of supervised models, whereas the second group consists of ICL models, *i.e.*, GPT-NeoX with a history length of 100. The best model for each dataset in the first group is shown in **bold**, and the second best is underlined.

| Single-Step | ICEWS14 | | | ICEWS18 | | |
|---|---|---|---|---|---|---|
| | H@1 | H@3 | H@10 | H@1 | H@3 | H@10 |
| frequency | 0.243 | 0.387 | 0.532 | 0.141 | 0.265 | 0.409 |
| recency | 0.228 | 0.387 | 0.536 | 0.120 | 0.242 | 0.403 |
| GPT-NeoX (Entity) | **0.324** | **0.460** | **0.565** | 0.192 | **0.313** | **0.414** |
| GPT-NeoX (Pair) | 0.297 | 0.408 | 0.482 | **0.196** | 0.307 | 0.402 |

(a) Single-step

| Multi-Step | ICEWS14 | | | ICEWS18 | | |
|---|---|---|---|---|---|---|
| | H@1 | H@3 | H@10 | H@1 | H@3 | H@10 |
| frequency | 0.222 | 0.349 | 0.460 | 0.121 | 0.207 | 0.307 |
| recency | 0.151 | 0.268 | 0.423 | 0.074 | 0.149 | 0.266 |
| GPT-NeoX (Entity) | **0.247** | **0.363** | **0.471** | 0.136 | 0.224 | 0.321 |
| GPT-NeoX (Pair) | 0.236 | 0.324 | 0.395 | **0.155** | **0.245** | **0.331** |

(b) Multi-step

Table 6: **Performance (Hits@K)** with rule-based predictions. The best model for each dataset is shown in **bold**.

## 4.4 ICL Implementation Details.

We implement our frameworks using PyTorch (Paszke et al., 2019) and Huggingface (Wolf et al., 2020). We first collate the facts $f \in \mathcal{D}_{test}$ based on the identical test query to eliminate any repeated inference. To illustrate, suppose there exist two facts in the test set denoted as $(s, p, a, t)$ and $(s, p, b, t)$ in the object prediction scenario. We consolidate these facts into $(s, p, [a, b], t)$ and forecast only one for $(s, p, ?, t)$. Subsequently, we proceed to generate an output for each test query with history by utilizing the model, obtaining the probability for the first generated token in a greedy approach, and sorting the probability. The outputs are deterministic for every iteration. We retain the numerical tokens corresponding to the numerical label $n$ that was targeted, selected from the top 100 probability tokens for each test query. To facilitate multi-step prediction, we incorporate the top-$k$ predictions of each test query as supplementary reference history. In this paper, we present results with $k = 1$. It is important to acknowledge that the prediction may contain minimal or no numerical tokens as a result of inadequate in-context learning. This can lead to problems when evaluating rank-based metrics. To mitigate this, we have established a protocol where

the rank of the actual value within the predictions is assigned a value of 100, which is considered incorrect according to our evaluation metric. For instruction-tuned model, we use the manual curated system instructions in Appendix A.3.

## 5 Experimental Results

### 5.1 In-context learning for TKG Forecasting

In this section, we present a multifaceted performance analysis of ICL under Index & Unidirection prompt strategy for both Entity and Pair history.

**Q1: How well does ICL fare against the supervised learning approaches?** We present a comparative analysis of the top-performing ICL model against established supervised learning methodologies for TKG reasoning, which are mostly based on graph representation learning. As evident from the results in Table 5, GPT-NeoX with a history length of 100 shows comparable performance to supervised learning approaches, without any fine-tuning on any TKG training dataset.

**Q2: How do frequency and recency biases affect ICL's predictions?** To determine the extent to which LLMs engage in pattern analysis, beyond simply relying on frequency and recency biases,

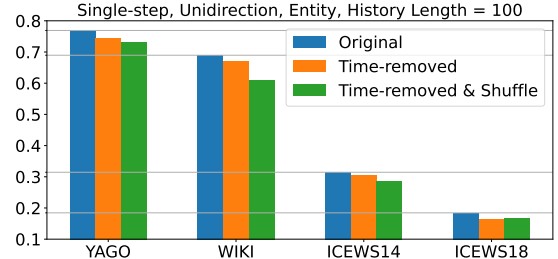

Figure 1: **Performance (Hit@1)** with and without time, and with shuffling

we run a comparative analysis between GPT-NeoX and heuristic-rules (*i.e.,* frequency & recency) on the ICEWS14 dataset, with history length set to 100. frequency identifies the target that appears most frequently in the provided history while recency selects the target associated with the most recent fact in the provided history. The reason for our focus on ICEWS is that each quadruple represents a single distinct event in time. In contrast, the process of constructing YAGO and WIKI involves converting durations to two timestamps to display events across the timeline. This step has resulted in recency heuristics outperforming all of the existing models, showcasing the shortcoming of existing TKG benchmarks (See Appendix A.5). The experimental results presented in Table 6 demonstrate that ICL exhibits superior performance to rule-based baselines. This finding suggests that ICL does not solely rely on specific biases to make predictions, but rather it actually learns more sophisticated patterns from historical data.

**Q3: How does ICL use the sequential and temporal information of events?** To assess the ability of LLMs to comprehend the temporal information of historical events, we compare the performance of prompts with and without timestamps. Specifically, we utilize the original prompt format, "$f_t$:[$\mathcal{I}(f_s)$, $\mathcal{I}(f_r), n_{f_o}. \mathcal{I}(f_o)$]", and the time-removed prompt format, "[$\mathcal{I}(f_s), \mathcal{I}(f_r), n_{f_o}. \mathcal{I}(f_o)$]", make the comparison (See Appendix A.2). Additionally, we shuffle the historical facts in the time-removed prompt format to see how the model is affected by the corruption of sequential information. Figure 1 shows that the absence of time reference can lead to a deterioration in performance, while the random arrangement of historical events may further exacerbate this decline in performance. This observation implies that the model has the capability to forecast the subsequent event by comprehending the sequential order of events.

| Single-Step | Prompt | ICEWS14 |
| --- | --- | --- |
| | | H@1 |
| gpt-3.5-turbo | index | 0.1615 |
| gpt-3.5-turbo | lexical | **0.1858** |

Table 7: **Performance (Hits@1)** between index and lexical for gpt-3.5-turbo.

**Q4: How does instruction-tuning affect ICL's performance?** To investigate the impact of instruction-tuning on ICL, we employ the gpt-3.5-turbo model with manually curated system instruction detailed in Appendix 4.4. Since the size of this model is not publicly disclosed, it is challenging to make direct comparisons with other models featured in this paper. Moreover, since this model does not provide output probabilities, we are only able to report the Hit@1 metric.

Table 7 showcases that the performance of the lexical prompts exceeds that of the index prompts by 0.024, suggesting that instruction-tuned models can make better use of semantic priors. This behavior is different from the other foundation LLMs, where the performance gap between the two prompt types was insignificant (See Figure 4 (a)).

**Q5: How does history length affect ICL's performance?** To evaluate the impact of the history length provided in the prompt, we conduct a set of experiments using varying history lengths. For this purpose, we use the best performing prompt format for each benchmark, *i.e.,* Entity for WIKI, YAGO, ICEWS18, and Pair for ICEWS14. Our results, as shown in Figure 2, indicate a consistent improvement in performance as the history length increases. This suggests that the models learn better as additional historical facts are presented. This observation is connected to few-shot learning in other domains, where performance improves as the number of examples per label increases. However, in our case, the historical patterns presented in the prompt do not explicitly depict the input-label mapping but rather aid in inferring the next step.

**Q6: What is the relation between ICL's performance and model size?** Here, we analyze the connection between model size and performance. Our results, as presented in Figure 3, conform to the expected trend of better performance with larger models. This finding aligns with prior works showing the scaling law of in-context learning performance. Our findings are still noteworthy since they

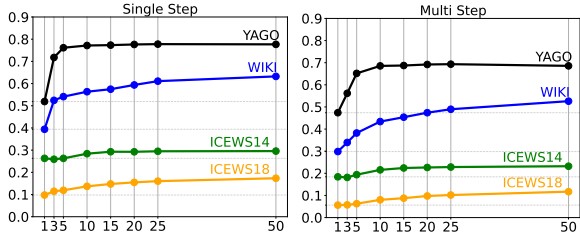

Figure 2: **Performance (Hit@1)** adheres to the scaling law based on the history length.

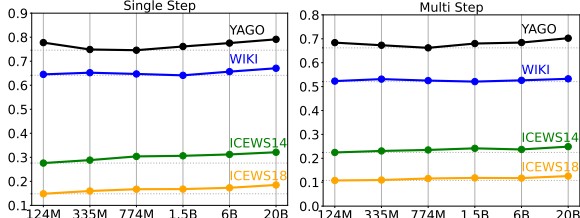

Figure 3: **Performance (Hit@1)** adheres to the scaling law based on the model size.

show how scaling model size can facilitate more powerful pattern inference for forecasting tasks.

## 5.2 Prompt Construction for TKG Forecasting

To determine the most effective prompt variation, we run a set of experiments on all prompt variations, using GPT-J (Wang, 2021) and under the single-step setting. Comprehensive results for prompt variations can be found in Appendix A.5.

**Index vs. Lexical**    Our first analysis compares the performance of `index` and `lexical` prompts. This investigation aims to determine whether the model relies solely on input-label mappings or if it also incorporates semantic priors from pre-training to make predictions. Our results (Figure 4 (a)) show that the performance is almost similar ($\pm 4e-3$ on average) across the datasets. This finding is aligned with previous studies indicating that foundation models depend more on input-label mappings and are minimally impacted by semantic priors (Wei et al., 2023).

**Unidirectional vs. Bidirectional**    We next analyze how the relation direction in the history modeling impacts the performance. This analysis aims to ascertain whether including historical facts, where the query entity or pair appears in any position, can improve performance by offering a diverse array of historical facts. Our results (Figure 4 (b)) show that there is a slight decrease in performance when `Bidirectional` history is employed, with a significant drop in performance observed particularly in the ICEWS benchmarks. These observations may be attributed to the considerably more significant number of entities placed in both subject and object positions in ICEWS benchmarks than YAGO and WIKI benchmarks (See Appendix A.5). This finding highlights the necessity of having robust constraints on the historical data for ICL to comprehend the existing pattern better.

**Entity vs. Pair**    Finally, we examine the impact of the history retrieval query on performance. Our hypothesis posits that when the query is limited to a single entity, we can incorporate more diverse historical facts. Conversely, when the query is a pair, we can acquire a more focused set of historical facts related to the query. Our results (Figure 4 (c)) indicate that the performance of the model is dependent on the type of data being processed. Specifically, the WIKI and ICEWS18 benchmarks perform better when the query is focused on the entity, as a broader range of historical facts is available. In contrast, the ICEWS14 benchmark performs better when the query is focused on pairs, as the historical facts present a more focused pattern.

## 6 Related Works

**Event Forecasting.**    Forecasting is a complex task that plays a crucial role in decision-making and safety across various domains (Hendrycks et al., 2021). To tackle this challenging task, researchers have explored various approaches, including *statistical* and *judgmental forecasting* (Webby and O'Connor, 1996; Armstrong, 2001; Zou et al., 2022). *Statistical forecasting* involves leveraging probabilistic models (Hyndman and Khandakar, 2008) or neural networks (Li et al., 2018; Sen et al., 2019) to predict trends over time-series data. While this method works well when there are many past observations and minimal distribution shifts, it is limited to numerical data and may not capture the underlying causal factors and dependencies that affect the outcome. On the other hand, *judgmental forecasting* involves utilizing diverse sources of information, such as news articles and external knowledge bases, to reason and predict future events. Recent works have leveraged language models to enhance reasoning capabilities when analyzing unstructured text data to answer forecasting inquiries (Zou et al., 2022; Jin et al., 2021).

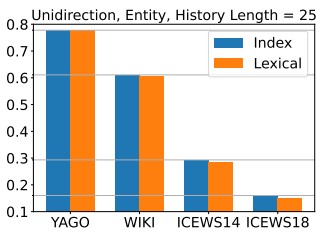 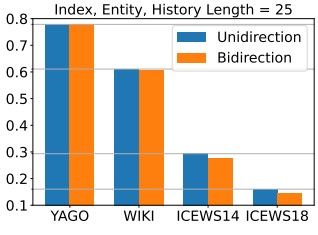 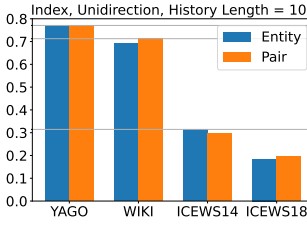

| (a) Index vs. Lexical | (b) Unidirection vs. Bidirection | (c) Entity vs. Pair |

Figure 4: **Performance (Hit@1) Analysis on Prompt Variation.** The comparable performance exhibited by both the Index and Lexical models indicates that these models rely heavily on learning patterns and are less influenced by semantic priors. Moreover, the Unidirectional model typically outperforms the Bidirectional model, suggesting that the robust constraints on historical data enable the model to comprehend observed patterns better. Finally, the performance of the Entity and Pair models varies depending on the dataset.

**Temporal Knowledge Graph.** Temporal knowledge graph (TKG) reasoning models are commonly employed in two distinct settings, namely *interpolation* and *extrapolation*, based on the facts available from $t_0$ to $t_n$. (1) **Interpolation** aims to predict missing facts within this time range from $t_0$ to $t_n$, and recent works have utilized embedding-based algorithms to learn low-dimensional representations for entities and relations to score candidate facts (Leblay and Chekol, 2018; García-Durán et al., 2018; Goel et al., 2020; Lacroix et al., 2020); (2) **Extrapolation** aims to predict future facts beyond $t_n$. Recent studies have treated TKGs as a sequence of snapshots, each containing facts corresponding to a timestamp $t_i$, and proposed solutions by modeling multi-relational interactions among entities and relations over these snapshots using graph neural networks (Jin et al., 2020; Li et al., 2021; Han et al., 2021b,a), reinforcement learning (Sun et al., 2021) or logical rules (Zhu et al., 2021; Liu et al., 2022). In our work, we focus on the *extrapolation* setting.

**In-context Learning.** In-context learning (ICL) has enabled LLMs to accomplish diverse tasks in a few-shot manner without needing parameter adjustments (Brown et al., 2020; Chowdhery et al., 2022). In order to effectively engage in ICL, models can leverage semantic prior knowledge to accurately predict labels following the structure of in-context exemplars (Min et al., 2022; Razeghi et al., 2022; Xie et al., 2022; Chan et al., 2022; Hahn and Goyal, 2023), and learn the input-label mappings from the in-context examples presented (Wei et al., 2023). To understand the mechanism of ICL, recent studies have explored the ICL capabilities of LLMs with regards to the impact of semantic prior knowl-

edge by examining their correlation with training examples (Min et al., 2022; Razeghi et al., 2022; Xie et al., 2022), data distribution (Chan et al., 2022), and language compositionality (Hahn and Goyal, 2023) in the pre-training corpus. Other recent works show that LLMs can actually learn input-label mappings from in-context examples by showing the transformer models trained on specific linear function class is actually predicting accurately on new unseen linear functions (Garg et al., 2022). More recently, there is a finding that large-enough models can still do ICL using input-label mappings when semantic prior knowledge is not available (Wei et al., 2023).

## 7 Conclusion

In this paper, we examined the forecasting capabilities of in-context learning in large language models. To this end, we experimented with temporal knowledge graph forecasting benchmarks. We presented a framework that converts relevant historical facts into prompts and generates ranked link predictions through token probabilities. Our experimental results demonstrated that without any fine-tuning and only through ICL, LLMs exhibit comparable performance to current supervised TKG methods that incorporate explicit modules to capture structural and temporal information. We also discovered that using numerical indices instead of entity/relation names does not significantly affect the performance, suggesting that prior semantic knowledge is not critical for overall performance. Additionally, our analysis indicated that ICL helps the model learn irregular patterns from historical facts, beyond simply making predictions based on the most common or the most recent facts in the given context. Together, our results and analyses

demonstrated that ICL can be a valuable tool for predicting future links using historical patterns, and also prompted further inquiry into the potential of ICL for additional capabilities.

## 8 Limitations

There are certain limitations to our experiments. First, computing resource constraints restrict our experiments to small-scale open-source models. Second, our methodologies have constraints regarding models where the tokenizer vocabulary comprises solely of single-digit numbers as tokens, such as LLAMA (Touvron et al., 2023). The performance of such models exhibits a similar trend in terms of scaling law concerning model size and history length, but these models demonstrate inferior performance compared to other models of the same model size. Third, our methodologies have certain limitations with respect to link prediction settings. While real-world forecasting can be performed in the transductive setting, where the answer can be an unseen history, our approach is constrained to the inductive setting, where the answer must be one of the histories observed. There are further directions that can be pursued. The first is to explore transductive extrapolation link prediction using LLMs. The second is to analyze the effects of fine-tuning on the results. Lastly, there is the opportunity to investigate the new capabilities of ICL.

## 9 Acknowledgement

This work was funded in part by the Defense Advanced Research Projects Agency (DARPA) and Army Research Office (ARO) under Contract No. W911NF-21-C-0002 and Contract No. HR00112290106, and with support from the Keston Exploratory Research Award and Amazon.

The views and conclusions contained herein are those of the authors and should not be interpreted as necessarily representing the official policies, either expressed or implied, of DARPA, ARO or the U.S. Government.

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

# A Appendix

## A.1 Prompt Example

Given the test query at timestamp 571, prompt examples for `Index` and `Lexical` are shown in Figure 5. Here, we assume the entity dictionary contains "Islamist Militia (Mozambique)" as index 0, "Meluco" as 10, "Namatil" as 36, "Muatide" as 53, "Limala" as 54, and "Nacate" as 55, while relation dictionary contains "Battles" as index 1 and "Violence against civilians" as 4. Also, the history setting is unidirectional entity setting where the history length is set to 5.

```
568: [0, 4, 1. 55]        568: [Islamist Militia (Mozambique), Violence against civilians, 1. Nacate]
568: [0, 4, 3. 10]        568: [Islamist Militia (Mozambique), Violence against civilians, 3. Meluco]
568: [0, 1, 2. 36]        568: [Islamist Militia (Mozambique), Battles, 2. Namatil]
569: [0, 1, 0. 54]        569: [Islamist Militia (Mozambique), Battles, 0. Limala]
570: [0, 1, 4. 53]        570: [Islamist Militia (Mozambique), Battles, 4. Muatide]
571: [0, 1,               571: [Islamist Militia (Mozambique), Battles,
        (a) Index                              (b) Lexical
```

Figure 5: **Prompt examples** for `Index` and `Lexical` settings.

## A.2 Prompt Example for Analysis

To assess the ability of LLMs to comprehend the sequential information of historical events, we compare the performance of prompts with and without timestamps (See Section 5.1 Q3). Figure 6 shows the prompt examples for time-removed and shuffled version of prompts.

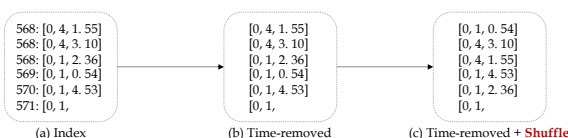

```
568: [0, 4, 1. 55]        [0, 4, 1. 55]         [0, 1, 0. 54]
568: [0, 4, 3. 10]        [0, 4, 3. 10]         [0, 4, 3. 10]
568: [0, 1, 2. 36]        [0, 1, 2. 36]         [0, 4, 1. 55]
569: [0, 1, 0. 54]        [0, 1, 0. 54]         [0, 1, 4. 53]
570: [0, 1, 4. 53]        [0, 1, 4. 53]         [0, 1, 2. 36]
571: [0, 1,               [0, 1,                [0, 1,
       (a) Index          (b) Time-removed      (c) Time-removed + Shuffle
```

Figure 6: **Prompt examples** for `time-removed` and `shuffled` version.

## A.3 System Instruction for Instruction-tuned models.

For the instruction-model, we use the manual curated system instructions to provide task descriptions and constraint the output format as follow:

```
You must be able to correctly predict the next
{object_label} from a given text consisting of
multiple quadruplets in the form of
"{time}:[{subject}, {relation}, {object_label}.
{object}]" and the query in the form of
"{time}:[{subject}, {relation}," in the end.

You must generate only the single number for
{object_label} without any explanation.
```

## A.4 Baseline Models

**RE-Net (Jin et al., 2020)** leverages an autoregressive architecture that employs a two-step process for learning temporal dependency from a sequence of graphs and local structural dependency from the vicinity. The model represents the likelihood of a fact occurring as a probability distribution that is conditioned on the sequential history of past snapshots.

**RE-GCN (Li et al., 2021)** also employs autoregressive architecture while it utilizes multi-layer relation-aware GCN on each graph snapshot to capture the structural dependencies among concurrent facts. Furthermore, the static properties of entities such as entity types, are also incorporated via a static graph constraint component to obtain better entity representations.

**TANGO (Han et al., 2021b)** employs autoregressive architecture as well but the use of continuous-time embedding in encoding temporal and structural information is a distinguishing feature of the proposed method, as opposed to RE-Net (Jin et al., 2020) (Li et al., 2021) and RE-GCN which operate on a discrete level with regards to time.

**xERTE (Han et al., 2021a)** employs an attention mechanism that can effectively capture the relevance of important aspects by selectively focusing on them. It employs a sequential reasoning approach over local subgraphs. This process begins with the query and iteratively selects relevant edges of entities within the subgraph, subsequently propagating attention along these edges. After multiple rounds of expansion, the final subgraph represents the interpretable reasoning path towards the predicted outcomes.

**TimeTraveler (Sun et al., 2021)** employs reinforcement learning for forecasting. The approach involves the use of an agent that navigates through historical knowledge graph snapshots, commencing from the query subject node. Thereafter, it sequentially moves to a new node by leveraging temporal facts that are linked to the current node, with the ultimate objective of halting at the answer node. To accommodate the issue of unseen-timestamp, the approach incorporates a relative time encoding function that captures time-related information when making decisions.

**CyGNet (Zhu et al., 2021)** leverages the statistical relevance of historical facts, acknowledging the recurrence of events in the temporal knowledge graph datasets. It incorporates two inference modes, namely Copy and Generation. The Copy mode determines the likelihood of the query being a repetition of relevant past facts. On the other hand, the Generation mode estimates the probability of each potential candidate being the correct prediction, using a linear classifier. The final forecast is obtained by aggregating the outputs of both modes.

**TLogic (Liu et al., 2022)** mines cyclic temporal logical rules by extracting temporal random walks from a graph. This process involves the extraction of temporal walks from the graph, followed by a lift to a more abstract, semantic level, resulting in the derivation of temporal rules that can generalize to new data. Subsequently, the application of these rules generates answer candidates, with the body groundings in the graph serving as explicit and easily comprehensible explanations for the results obtained.

### A.5   Full Experimental Results

| Prompt | History | YAGO | | | WIKI | | | ICEWS14 | | | ICEWS18 | | | ACLED | | |
|---|---|---|---|---|---|---|---|---|---|---|---|---|---|---|---|---|
| | | H@1 | H@3 | H@10 | H@1 | H@3 | H@10 | H@1 | H@3 | H@10 | H@1 | H@3 | H@10 | H@1 | H@3 | H@10 |
| Index | Unidirectional Entity | 0.777 | 0.880 | 0.904 | 0.610 | 0.724 | 0.775 | 0.293 | 0.427 | 0.533 | 0.160 | 0.267 | 0.395 | 0.364 | 0.497 | 0.613 |
| Index | Unidirectional Pair | 0.778 | 0.880 | 0.904 | 0.646 | 0.731 | 0.777 | 0.294 | 0.400 | 0.471 | 0.187 | 0.294 | 0.385 | 0.331 | 0.457 | 0.564 |
| Index | Bidirectional Entity | 0.778 | 0.879 | 0.904 | 0.607 | 0.721 | 0.773 | 0.274 | 0.404 | 0.527 | 0.142 | 0.245 | 0.382 | 0.336 | 0.497 | 0.613 |
| Index | Bidirectional Pair | 0.780 | 0.879 | 0.904 | 0.647 | 0.733 | 0.777 | 0.291 | 0.398 | 0.471 | 0.185 | 0.291 | 0.384 | 0.324 | 0.455 | 0.564 |
| Lexical | Unidirectional Entity | 0.777 | 0.874 | 0.904 | 0.607 | 0.722 | 0.775 | 0.285 | 0.406 | 0.532 | 0.149 | 0.257 | 0.394 | 0.293 | 0.469 | 0.601 |
| Lexical | Unidirectional Pair | 0.781 | 0.874 | 0.904 | 0.645 | 0.730 | 0.777 | 0.280 | 0.392 | 0.470 | 0.171 | 0.278 | 0.381 | 0.277 | 0.426 | 0.549 |
| Lexical | Bidirectional Entity | 0.773 | 0.872 | 0.904 | 0.601 | 0.714 | 0.771 | 0.270 | 0.399 | 0.526 | 0.141 | 0.238 | 0.371 | 0.343 | 0.464 | 0.601 |
| Lexical | Bidirectional Pair | 0.777 | 0.874 | 0.904 | 0.643 | 0.728 | 0.777 | 0.278 | 0.390 | 0.470 | 0.182 | 0.288 | 0.381 | 0.293 | 0.426 | 0.554 |

Table 8: **Performance Comparison (Hits@k)** in single-step inference with time-aware filter.

| Prompt | History | YAGO | | | WIKI | | | ICEWS14 | | | ICEWS18 | | | ACLED | | |
|---|---|---|---|---|---|---|---|---|---|---|---|---|---|---|---|---|
| | | H@1 | H@3 | H@10 | H@1 | H@3 | H@10 | H@1 | H@3 | H@10 | H@1 | H@3 | H@10 | H@1 | H@3 | H@10 |
| Index | Unidirectional Entity | 0.693 | 0.790 | 0.820 | 0.489 | 0.555 | 0.587 | 0.209 | 0.303 | 0.377 | 0.101 | 0.161 | 0.223 | 0.348 | 0.433 | 0.478 |
| Index | Unidirectional Pair | 0.694 | 0.790 | 0.820 | 0.518 | 0.567 | 0.596 | 0.228 | 0.315 | 0.381 | 0.143 | 0.224 | 0.298 | 0.317 | 0.400 | 0.438 |
| Index | Bidirectional Entity | 0.692 | 0.970 | 0.820 | 0.483 | 0.548 | 0.580 | 0.174 | 0.247 | 0.318 | 0.082 | 0.126 | 0.178 | 0.312 | 0.433 | 0.478 |
| Index | Bidirectional Pair | 0.694 | 0.790 | 0.820 | 0.518 | 0.567 | 0.595 | 0.226 | 0.313 | 0.379 | 0.142 | 0.223 | 0.298 | 0.308 | 0.400 | 0.438 |
| Lexical | Unidirectional Entity | 0.697 | 0.789 | 0.820 | 0.487 | 0.557 | 0.588 | 0.218 | 0.306 | 0.379 | 0.102 | 0.161 | 0.223 | 0.341 | 0.428 | 0.481 |
| Lexical | Unidirectional Pair | 0.698 | 0.789 | 0.820 | 0.524 | 0.570 | 0.597 | 0.230 | 0.317 | 0.382 | 0.143 | 0.226 | 0.299 | 0.303 | 0.393 | 0.445 |
| Lexical | Bidirectional Entity | 0.693 | 0.789 | 0.820 | 0.479 | 0.547 | 0.579 | 0.178 | 0.255 | 0.327 | 0.091 | 0.122 | 0.168 | 0.338 | 0.428 | 0.481 |
| Lexical | Bidirectional Pair | 0.697 | 0.789 | 0.820 | 0.521 | 0.568 | 0.596 | 0.227 | 0.315 | 0.379 | 0.141 | 0.212 | 0.287 | 0.312 | 0.393 | 0.445 |

Table 9: **Performance Comparison (Hits@k)** in multi-step inference with time-aware filter.

| Single-Step | YAGO | | | WIKI | | | ICEWS14 | | | ICEWS18 | | |
|---|---|---|---|---|---|---|---|---|---|---|---|---|
| | H@1 | H@3 | H@10 | H@1 | H@3 | H@10 | H@1 | H@3 | H@10 | H@1 | H@3 | H@10 |
| frequency | 0.766 | 0.859 | 0.921 | 0.549 | 0.712 | 0.818 | 0.243 | 0.387 | 0.532 | 0.141 | 0.265 | 0.409 |
| recency | 0.886 | 0.927 | 0.928 | 0.701 | 0.831 | 0.849 | 0.228 | 0.387 | 0.536 | 0.120 | 0.242 | 0.403 |
| GPT-NeoX (Entity) | 0.784 | 0.891 | **0.927** | 0.694 | 0.804 | 0.844 | **0.324** | **0.460** | **0.565** | 0.192 | **0.313** | **0.414** |
| GPT-NeoX (Pair) | **0.787** | **0.892** | 0.926 | **0.721** | **0.812** | **0.847** | 0.297 | 0.408 | 0.482 | **0.196** | 0.307 | 0.402 |

Table 10: **Performance Comparison (Hits@k)** between rule-based prediction and ICL in single-step inference with time-aware filter.

| Single-Step | YAGO | | | WIKI | | | ICEWS14 | | | ICEWS18 | | |
|---|---|---|---|---|---|---|---|---|---|---|---|---|
| | H@1 | H@3 | H@10 | H@1 | H@3 | H@10 | H@1 | H@3 | H@10 | H@1 | H@3 | H@10 |
| frequency | 0.691 | 0.789 | 0.837 | 0.484 | 0.603 | 0.652 | 0.222 | 0.349 | 0.460 | 0.121 | 0.207 | 0.307 |
| recency | 0.785 | 0.840 | 0.842 | 0.540 | 0.637 | 0.661 | 0.151 | 0.268 | 0.423 | 0.074 | 0.149 | 0.266 |
| GPT-NeoX (Entity) | 0.686 | **0.793** | **0.840** | 0.543 | 0.622 | **0.655** | **0.247** | **0.363** | **0.471** | 0.136 | 0.224 | 0.321 |
| GPT-NeoX (Pair) | **0.688** | **0.793** | 0.839 | **0.570** | **0.625** | 0.652 | 0.236 | 0.324 | 0.395 | **0.155** | **0.245** | **0.331** |

Table 11: **Performance Comparison (Hits@k)** between rule-based prediction and ICL in multi-step inference with time-aware filter.