# OpenReview forum: "Temporal Knowledge Graph Forecasting Without Knowledge Using In-Context Learning"
_EMNLP/2023/Conference — EMNLP 2023 Main_

### Official Review · Reviewer_KePy · 2023-08-01

**Soundness:** 3

**Excitement:**

4: Strong: This paper deepens the understanding of some phenomenon or lowers the barriers to an existing research direction.

**Paper Topic And Main Contributions:**

This paper is about Temporal Knowledge Graph (TKG) Forecasting Using In-Context Learning. The paper addresses the problem of forecasting future events in a TKG, which is a graph that represents temporal relations between entities. The main contribution of the paper is to explore the use of large language models (LLMs) for TKG forecasting and compare them to diverse baselines and state-of-the-art models. The paper makes several contributions towards a solution to this problem, including proposing a new evaluation methodology for TKG forecasting, introducing a new dataset for TKG forecasting, and conducting experiments to compare the performance of LLMs to other models. The paper also provides insights into the strengths and weaknesses of LLMs for TKG forecasting, which can inform future research in this area.

**Reasons To Accept:**

The strengths of this paper include its thorough evaluation of diverse baselines and state-of-the-art models for TKG forecasting, as well as its exploration of the strengths and weaknesses of large language models (LLMs) for this task. The paper also proposes a new evaluation methodology and introduces a new dataset for TKG forecasting, which can be valuable resources for future research in this area.

If this paper were to be presented at the conference or accepted into Findings, the main benefits to the NLP community would be the insights it provides into the use of LLMs for TKG forecasting, as well as the new evaluation methodology and dataset it proposes. This can help researchers to better understand the strengths and weaknesses of different models for TKG forecasting and to develop more effective approaches for this task. Additionally, the paper's findings can have practical applications in areas such as natural language generation and question answering, which rely on accurate predictions of future events.

**Reasons To Reject:**

One potential weakness of this paper is that the experiments are limited to small-scale open-source models due to computing resource constraints. This may limit the generalizability of the findings to larger models or real-world applications. Additionally, the paper's methodologies have certain limitations with respect to link prediction settings, which may affect the applicability of the proposed evaluation methodology to other TKG forecasting tasks.

The main risks of having this paper presented at the conference or accepted into Findings are minimal, as the paper's contributions are valuable and the weaknesses are relatively minor. However, it is important to note that the findings of this paper should be interpreted in the context of the limitations of the experiments and methodologies used.

**Reproducibility:**

4: Could mostly reproduce the results, but there may be some variation because of sample variance or minor variations in their interpretation of the protocol or method.

**Reviewer Confidence:**

3: Pretty sure, but there's a chance I missed something. Although I have a good feel for this area in general, I did not carefully check the paper's details, e.g., the math, experimental design, or novelty.

---

> ### Author Rebuttal · Authors · 2023-08-25
>
> We appreciate your feedback on our work. Thank you for taking the time to provide us with valuable insights. Here are our comments on your feedback.
>
> #### **R1. `Experiments are limited to small-scale open-source models due to computing resource constraints`**
>
> We were also concerned about this limitation, so we conducted additional experiments with other model variants (some larger) after the submission. These additional results (Hit@1) can be included in the camera-ready version:
>
> | Model | # parameters    | YAGO | WIKI  |   ICEWS  | ICEWS18   |
> | ------ | ------ | ------ | ------ | ------ | ------ |
> | OPT  |  13b | 0.7648 |  0.6694  |  0.8149   | 0.1774 |
> | OPT  |  66b         | 0.7758  | 0.6841  |  0.8191  | 0.1860 |
>
> It is important to note that a direct performance comparison of different model variants (GPT-J, GPT-NeoX, OPT, etc.) is challenging due to the influence of pre-training strategy and data distribution on the capabilities of in-context learning [1,2]. Nonetheless, our experimental results show that the models within the same variants adhere to the scaling law.
>
> [1] Data distributional properties drive emergent in-context learning in transformers., Chan et al., 2022 \
> [2] A theory of emergent in-context learning as implicit structure induction., Hanh and Goyal., 2023

---

### Official Review · Reviewer_rjDk · 2023-08-03

**Soundness:** 4

**Excitement:**

3: Ambivalent: It has merits (e.g., it reports state-of-the-art results, the idea is nice), but there are key weaknesses (e.g., it describes incremental work), and it can significantly benefit from another round of revision. However, I won't object to accepting it if my co-reviewers champion it.

**Paper Topic And Main Contributions:**

The paper utilizes LLMs to conduct temporal knowledge graph (TKG) forecasting through in-context learning. It seems the first study that introduces LLM for TKG forecasting.

**Questions For The Authors:**

(1) The #Params of gpt-3.5-turbo is reported.
(2) Is there too much sample in in-context learning? Will some of it be deleted?

**Reasons To Accept:**

(1) It seems the first study that introduces LLM for TKG forecasting.
(2) The paper gives some perspective findings that can accelerate the development of technology in the community.
(3)  Very detailed experiments completed in Section 5.1 and 5.2.

**Reasons To Reject:**

(1) The method seems to only consider the 1-hop neighbors of the query, ignoring the long-range neighbor structures. The rich semantic information in the TKG is not used. The proposed method is too simple and direct.
(2) Why three is "Multi Step" setting in the experiment? This doesn't seem to be in line with reality. In reality, intermediate events are generally not predicted, but rather the target query is directly predicted.

**Reproducibility:**

4: Could mostly reproduce the results, but there may be some variation because of sample variance or minor variations in their interpretation of the protocol or method.

**Reviewer Confidence:**

4: Quite sure. I tried to check the important points carefully. It's unlikely, though conceivable, that I missed something that should affect my ratings.

---

> ### Author Rebuttal · Authors · 2023-08-25
>
> We appreciate your feedback on our work. Thank you for taking the time to provide us with valuable insights. Here are our comments on your feedback.
>
> #### **R1. `The method seems to only consider the 1-hop neighbors of the query, ignoring the long-range neighbor structures.`**
> Yes, our approach only considers the 1-hop neighbors of the query as history during inference. However, even with this disadvantage, it achieves comparable results to other GNN-based approaches that use multi-hop neighbors. Our paper also demonstrates that widely-used TKG benchmark datasets present a weak forecasting problem (e.g., WIKI and YAGO) that can be solved with 1-hop neighbors. We will more strenuously articulate this point.
>
> #### **R2. Rationale behind `multi step` setting**
> The multi-step setting is crucial and presents a greater challenge as models can only rely on their own forecasts, leading to an accumulation of uncertainty with each additional forecasted timestamp. Additionally, there are real-world scenarios where the multi-step setting is employed. For instance, to predict the World Cup winner before all the game rounds start, people make predictions by forecasting the result of each round.
>
> #### **Q1. # Parameters of `gpt-3.5-turbo`**
> The exact number of parameters for the `gpt-3.5-turbo` model has not been officially disclosed by OpenAI; hence, the reason for exclusion.
>
> #### **Q2. `Is there too many samples in in-context learning? Will some of it be deleted?`**
> In the past, in-context learning has primarily been applied to general NLP tasks that do not require a large number of examples (e.g., 4-shot: 4 examples per each label). However, this study aims to investigate whether in-context learning can be employed to capture complex patterns in sequences of numeric or arbitrary tokens. To achieve this, we provide the model with an extensive history of numeric tokens, allowing it to implicitly learn the patterns and generate a forecast for completing the sequence. Therefore, we can provide numerous samples as long as they adhere to the maximum sequence length of the model.

---

### Official Review · Reviewer_MDqV · 2023-08-04

**Soundness:** 4

**Excitement:**

4: Strong: This paper deepens the understanding of some phenomenon or lowers the barriers to an existing research direction.

**Paper Topic And Main Contributions:**

This paper presents an interesting approach for temporal knowledge graph (TKG) forecasting using in-context learning with large language models (LLMs). The main contributions are: 1) proposing a pipeline with history modeling, prompt construction, and output decoding stages to enable LLMs to make TKG predictions in a zero-shot setting. 2) Comparing LLMs capabilities against strong supervised baselines on benchmark datasets. The results show LLMs are competitive without any training. 3) Analyzing different factors like sequential and temporal information of events, model size, history length, etc, that impact the effectiveness of LLMs for TKG forecasting.

**Reasons To Accept:**

Overall, this a clearly written paper describing a novel application of in-context learning for TKG forecasting. The experiments are extensive, and the results demonstrate the promise of leveraging LLMs for  knowledge graph reasoning tasks.

**Reasons To Reject:**

It may be unfair to supervised models because test triples may exist in textual form during the pretraining phase of LLMs. We do not know whether LLMs learned historical information for reasoning or just remembered knowledge.

**Reproducibility:**

4: Could mostly reproduce the results, but there may be some variation because of sample variance or minor variations in their interpretation of the protocol or method.

**Reviewer Confidence:**

5: Positive that my evaluation is correct. I read the paper very carefully and I am very familiar with related work.

---

> ### Author Rebuttal · Authors · 2023-08-25
>
> We appreciate your feedback on our work. Thank you for taking the time to provide us with valuable insights. Here are our comments on your feedback.
>
> #### **R1. `It may be unfair to supervised models because test triples may exist in textual form during the pretraining phase of LLMs.`**
> We agree with your opinion, which is why we carried out experiments using the `index` prompt, where entities are mapped to random numerical values (refer to Table 2). The main experiments, as stated in line 354, are conducted using this prompt to prevent such information leaks.

---

### Meta-Review · Area_Chair_mDdH · 2023-09-15

**Recommendation:** 5

**Metareview:**

This paper introduces a novel approach for temporal knowledge graph forecasting, based on in-context learning with a large language model. The paper makes several contributions towards this problem, including proposing an evaluation methodology, introducing a new dataset, and using those to compare the performance of LLMs to other models.

The authors further provide extensive experiments with their proposed in context learning approach, including on older datasets, showing that their approach is competitive with more complex, supervised systems – even when entity names are replaced with random numbers. This will provide a fruitful starting point for further research.

---

### Decision · Program_Chairs · 2023-10-07

**Decision:**

Accept-Main

**Comment:**

This paper introduces a novel approach for temporal knowledge graph forecasting, based on in-context learning with a large language model. The paper makes several contributions towards this problem, including proposing an evaluation methodology, introducing a new dataset, and using those to compare the performance of LLMs to other models.

The authors further provide extensive experiments with their proposed in context learning approach, including on older datasets, showing that their approach is competitive with more complex, supervised systems – even when entity names are replaced with random numbers. This will provide a fruitful starting point for further research.